# Rational Drug and Antibiotic Use Status, E-Health Literacy in Syrian Immigrants and Related Factors: A Cross-Sectional Study

**DOI:** 10.3390/antibiotics12101531

**Published:** 2023-10-11

**Authors:** Mehmet Sait Değer, Mehmet Akif Sezerol, Muhammed Atak

**Affiliations:** 1Department of Public Health, Medical Faculty, Hitit University, Corum 19030, Türkiye; mehmetsaitdeger@hitit.edu.tr; 2Epidemiology Program, Institute of Health Sciences, Istanbul Medipol University, Istanbul 34810, Türkiye; masezerol@gmail.com; 3Sultanbeyli District Health Directorate, Istanbul 34935, Türkiye; 4Health Management Program, Graduate Education Institute, Maltepe University, Istanbul 34857, Türkiye; 5Department of Public Health, School of Medicine, Istanbul Medipol University, Istanbul 34810, Türkiye; 6Department of Public Health, Istanbul Medical Faculty, Istanbul University, Istanbul 34093, Türkiye

**Keywords:** immigrant, rational drug use, rational antibiotic use, eHealth literacy, eHEALS

## Abstract

Rational drug use is a pivotal concept linked with morbidity and mortality. Immigration plays a significant role as a determinant affecting individuals’ health-related attitudes, behaviors, and the pursuit of health services. Within this context, the study was initiated to assess the factors influencing health literacy and rational drug use among Syrian immigrants in Istanbul. A cross-sectional study was undertaken on 542 Syrian adults utilizing a three-part questionnaire encompassing sociodemographics, rational drug use, and the e-health literacy scale (eHEALS). With an average age of 39.19 ± 13.10 years, a majority of participants believed medications should solely be doctor-prescribed (97%) and opposed keeping antibiotics at home (93.7%). Yet, 62.5% thought excessive herbal medicine use was harmless. The mean eHEALS score stood at 20.57 ± 7.26, and factors like age, marital status, income, and duration of stay in Turkey influenced e-health literacy. Associations were seen between low e-health literacy and being female, being older, having a lower education level, and regular medication use. Syrian immigrants displayed proper knowledge concerning antibiotics yet exhibited gaps in their understanding of general drug usage, treatment adherence, and herbal medicines. Approximately 80.3% had limited health literacy, pointing to the need for targeted interventions for enhanced health and societal assimilation.

## 1. Introduction

Health is defined not only as the absence of disease and disability but also as a state of complete physical, mental and social well-being [1]. Psychosocial, economic and cultural factors and adequate utilization of health services are important in achieving and maintaining well-being [2]. The number of refugees and asylum seekers in the world is increasing. In 2022, 112.6 million people in the world were in the group defined as refugees or asylum seekers [3]. More than 3.5 million Syrian refugees live in Turkey. Health problems are more common in migrants [4]. In addition, the psychosocial and economic conditions of migrants and the language barriers they face negatively affect their health status as their search for health services remains limited [2]. Migration experience and cultural factors affect migrants’ perception of drugs and antibiotics, and unconscious drug use is common among migrants [5,6].

Antibiotic resistance is one of the most important global public health threats worldwide. In particular, unconscious and improper use of antibiotics accelerates the development of resistance, which affects the success of treatment of infectious diseases and the duration of hospitalization, leading to an increase in health-related costs and mortality rates [7,8]. Rational use of drugs, especially antibiotics, is an important factor that prevents morbidity and mortality related to diseases [9]. Inadequate health literacy, self-medication and over-the-counter medication supply are important factors leading to widespread and uncontrolled use of drugs [10]. Interventions in Turkey have shown that educational activities are effective in improving the prescription, distribution and utilization of antibiotics [11,12,13].

Health literacy is defined as the knowledge and cognitive and social competence required for individuals to access, understand, evaluate and use health-related information to protect and improve their health, make decisions about their health status and improve their quality of life [14,15]. Health literacy is an important public health goal that also refers to the state and competence of individuals to meet complex health needs [16,17,18]. Challenging living conditions, cultural factors, language barriers, the complex and multidimensional structure of the health system, and social and economic disadvantages negatively affect migrants’ search for and utilization of health services and their health in general [15]. The fact that information sources in health are diverse and information is dense has made the internet an important resource for accessing the right information for health. The internet is a useful and effective tool for accessing accurate health-related information and developing various skills to protect and improve health [19]. E-health literacy refers to an individual’s ability to search, find, understand and evaluate health-related information from digital sources and use it for any health condition and/or problem [1,18,19]. Various studies with migrants have shown that their health literacy levels (65.1–67.8%) are inadequate and problematic [20,21]. Immigration is an important social determinant of health related to access to health services, utilization of health services, health perception and health literacy [20,22]. Health literacy is of critical importance in eliminating health inequalities and increasing the health levels in society [21].

The health literacy levels of individuals is an important and determining factor in rational drug use. Therefore, efforts to increase the health literacy level of immigrants will contribute greatly to increasing their knowledge about rational drug use and developing positive attitudes. In this study, it was aimed to determine the rational drug use and health literacy levels of Syrian adults living in a district of Istanbul and to examine the related factors.

## 2. Results

The mean age and SD value of the research group was 39.19 ± 13.10. In this study, 52.2% (283 people) of the participants were female and 47.8% (259 people) were male. It was determined that 46.5% of the immigrants in the research group were in the age group of 40 and above, 76.9% were married, 53.0% had high school and higher education, 80.4% had low income, 64.0% had been living in Turkey for 7 years or more, 71.4% lived in the same house with 5 or more people, 36.5% had chronic diseases, 60.9% used regular medication and 87.1% applied to a physician in the first place when they got sick. Data on the sociodemographic characteristics and disease–health status of the research group are shown in Table 1.

In this study, 97.0% of the immigrants in the research group stated that medication should only be used when prescribed by a doctor. Furthermore, 93.7% stated that people should not keep antibiotics in their homes and then use them for other diseases. In total, 96.1% stated that physicians should prescribe antibiotics only when needed, and 75.8% stated that using enough medication, not too much, leads to recovery. Data on immigrants’ attitudes and approaches to rational drug use are shown in Table 2.

The eHEALS mean and SD value of the immigrants was 20.57 ± 7.26. It was determined that 80.3% of the immigrant group had limited health literacy, and 19.7% had adequate health literacy. Data on the eHealth Literacy Scale scores are shown in Table 3.

In the study group, the mean ranking of eHEALS was significantly higher in immigrants who were in the 30–39 age group, married, had low income, had been living in Turkey for 7 years or more, did not have chronic diseases, did not use regular medication and had a monthly out-of-pocket health expenditure of less than 500 TL (*p* ˂ 0.05). The comparison of sociodemographic and health-disease status with eHEALS is shown in Table 4.

Among the independent variables age group (*p*: 0.019 O.R: 2.83), gender (*p*: 0.048 O.R: 1.60), education level (*p*: 0.003 O.R: 3.96) and regular medication use (*p*: 0.000 O.R: 0.18) were found to contribute significantly to the model. The regression analysis of health literacy level according to sociodemographic characteristics is shown in Table 5.

## 3. Discussion

Socially and economically disadvantaged migrants are one of the groups that should be prioritized for public health interventions. It is important to determine the knowledge and behaviors of migrants regarding rational drug use. Health literacy is an important tool to increase the health level of individuals and society. The level of health literacy among migrants is a critically important determinant of rational drug use. In this study, we aimed to determine the rational drug use and e-health literacy levels of Syrian migrants and to evaluate the associated factors. It was found that 76.9% of the migrants in the study group were married, 53.0% had high school or higher education, 80.4% had low income, 60.9% used regular medication, 87.1% consulted a physician first when they got sick, and 80.3% had limited e-health literacy.

Approximately 3.5 million Syrian immigrants live in Turkey. The average age of immigrants is 22.32 years. Overall, 72.68% are women and children. In total, 30.23% are under the age of 10. Furthermore, 2.23% live in temporary shelter centers and 97.7% live in cities (Istanbul: 531.996, Gaziantep: 434.045, Şanlıurfa: 317.786), and the ratio of Syrian immigrants to Turkey’s population is 3.73% [23]. The fact that the individuals in the research group first consult a physician when they get sick shows that they care about their health and seek to protect their health. In addition, the 87.1% preference for consulting a physician when ill suggests that immigrants do not experience difficulties in accessing health services in Turkey. In a meta-analysis, it was shown that migration-related factors, as well as social and economic conditions, may affect the health of immigrants [24].

In this study, 97% of the immigrants in the research group stated that medication should only be used when prescribed by a doctor. Furthermore, 93.7% stated that people should not keep antibiotics in their homes and then use them for other illnesses. Moreover, 96.1% stated that doctors should prescribe antibiotics only when needed. In total, 75.8% stated that using enough medication, not too much, leads to recovery. In this study, 51.0% stated that people can stop taking medication if they feel well during treatment. In total, 38% stated that people can stop taking medication if they feel well during treatment, and 38% stated that people can stop taking medication if they feel well. In this study, 0% stated that there is no harm in recommending medication to their relatives with similar complaints. Of the sample, 38.7% stated that herbs can be used instead of medication. In this study, 62.5% stated that using herbal medication as much as desired is not harmful to health. Furthermore, 36.7% stated that the form and duration of medication use cannot be determined by the individual. In this study, 61.0% stated that medications cannot be used to the same extent in every age group. In total, 68.1% stated that the duration of use of medications is not the same, and 67.4% stated that expensive medications are not more effective. The fact that the majority of the immigrants in the study group have low income and about half of them have an education level below high school suggests that their knowledge and perceptions about antibiotics are not sufficient. However, the results of the study show that the general knowledge and perceptions of immigrants about antibiotic use are better than expected [25]. In addition, it is seen that medication compliance is low, and it is common to recommend medication to relatives with similar symptoms. On the other hand, the presence of positive perceptions of herbal medicines and their use among Syrian immigrants may be related to sociocultural factors and past experiences.

Increasing antibiotic resistance is now considered a public health problem because it poses both a threat to human health and a serious economic cost [26,27]. In a study conducted with immigrants in the Netherlands, it was shown that immigrants had a more limited perception and knowledge of antibiotics compared to the native population [5]. Although physical and mental health problems are common in immigrants, their low socioeconomic status is associated with poor health outcomes [28]. There are studies showing that treatment compliance is low in immigrants [29]. Studies conducted in Turkey have shown that age, marital status, education level, income level, family structure, place of residence, employment status and health education status are associated with rational drug use [30,31,32]. In a different study, it was shown that giving importance to health and seeking healthy life behaviors positively affected the attitude toward rational drug use [33]. In another study conducted in Turkey, sociodemographic characteristics such as age, gender, employment status and education level were found to be associated with the level of rational drug use knowledge of Syrian immigrants [34]. In a meta-analysis, it was shown that factors such as previous similar symptoms and antibiotic experiences, perceived low severity of the disease, intention to recover quickly, difficulty in accessing a physician or health facility, lack of trust, low cost and ease of use affect/increase self-medication [35]. In another meta-analysis, a positive relationship was found between health literacy and medication adherence [36].

Today, the internet has become a frequently used source of health information because of its ease of access and use, low cost and ubiquity. People frequently use the internet for disease prevention, healthy living behaviors and general disease conditions [37]. However, there may be some difficulties for users to access useful and quality health and medical information online [38]. It is inevitable that individuals with low income and low levels of e-health literacy, such as immigrants, will experience difficulties in this situation. As a matter of fact, the eHEALS median (min–max) value of the migrants in our research group was found to be 21 (8–34). The e-health literacy level of 80.3% of the immigrant group was found to be limited (insufficient + problematic), and 19.7% was found to be sufficient. In a recent study conducted in the same city, it was also observed that immigrant health literacy levels were insufficient [39]. Immigrants in the study group with low income levels may have limited internet access and use. In addition, low education level and sociocultural factors in the study group may have affected immigrants’ access to accurate and reliable information about health on the internet and their ability to understand and use this information. In addition, immigrants’ health perceptions, chronic disease status and health-information-seeking behaviors or habits may affect their e-health literacy levels. The level of education and health literacy of society affects the health status of individuals and their attitudes and perceptions towards medicines [40]. On the other hand, in disadvantaged groups such as immigrants and the elderly, technological applications can make a significant contribution to individuals’ access to reliable health information and making the right health decisions [41].

It is important that the health services of immigrant-hosting countries are appropriate to the personal needs, living conditions, sociocultural characteristics and competence levels of immigrants. Improving health literacy plays a critical role at this point. In Turkey, Syrian immigrants can access health services free of charge [42]. In addition, health services are provided to these immigrants by Syrian healthcare professionals through reinforced immigrant health centers. In these centers, where specialist physicians in various branches work, preventive health services (immunization, family planning, education, screening programs), outpatient diagnostic and therapeutic health services are provided without language barriers. This situation positively affects Syrian immigrants’ access to and use of health services and contributes to the protection and improvement of their health. It should not be overlooked that it also contributes positively to their health literacy status. A meta-analysis has shown that the concept of health literacy is very important for protecting and improving the health of individuals and is an important determinant of the health level of society [43]. Basic health literacy facilitates individuals’ access to health services, enables reducing health inequalities and contributes to the development of health services policies at the societal level [44].

In the research group, the mean ranks of e-health literacy were significantly higher in the age group of 30–39. Married, low income, living in Turkey for 7 years or more, not having chronic diseases, not using regular medication, not doing anything for a while when they get sick and using medication according to their own experience, and having a monthly out-of-pocket health expenditure of less than 500 TL (*p* ˂ 0.05). The presence of social support within the family among married individuals in the research group may have contributed to the well-being and better health of individuals. In a meta-analysis, the positive effect of education, income level and the presence of social support on individuals’ health literacy was shown [45]. Immigrants who live in Turkey for longer periods of time overcome the language barrier to a large extent. Since they are in contact with the community, their children or siblings go to school, and their spouses or family members work, there is always someone in the family who speaks Turkish. This makes it easier for Syrian immigrants who live in Turkey for longer periods of time to follow official procedures and access and use health services. As a matter of fact, Syrian immigrants who are registered in the city where they reside in Turkey can access public health services free of charge thanks to their temporary protection status, and they can also get their medicines free of charge or by paying co-payments. This is supported by the fact that the vast majority (84.1%) of immigrants in the study had a small out-of-pocket health expenditure (˂500 TL). The high level of eHealth literacy of immigrants who do not have chronic diseases and do not use regular medication may be related to the fact that they use digital resources more intensively in accessing reliable and accurate information about protecting their health and adopting healthy life behaviors because they care more about their health. Systematic reviews have shown that education level is associated with eHealth literacy [46]. It is inevitable for individuals with low health literacy to skip preventive health services, treatment compliance, chronic disease management and, more generally, have poor health outcomes [47]. Factors such as cultural beliefs about health and illness, language problems and socioeconomic status affect immigrants’ communication with healthcare providers and their understanding and compliance with medical instructions [48].

In the logistic regression analysis established to predict the level of eHealth literacy according to sociodemographic characteristics, model fit was found to be good. Since it is thought that the effect of some independent variables would be more significant within the scope of the research, these variables (age, gender, education level, chronic disease, continuous medication use, etc.) were included in the regression analysis. In addition, in order to obtain a stronger prediction model with fewer variables, regression analysis was performed only with some independent variables. Among the independent variables, age group (*p*: 0.019 O.R: 2.83), gender (*p*: 0.048 O.R: 1.60), education level (*p*: 0.003 O.R: 3.96) and regular medication use (*p*: 0.000 O.R: 0.18) were found to contribute significantly to the model. Female gender, advanced age, low education level and regular medication use decrease the level of health literacy. In the traditional sociocultural structure of Syrian immigrants, it is mostly men who have more contact with the outside social environment, attend school and have a job. For this reason, immigrant women are less likely to access the internet as they lack both language learning and economic independence. This situation also contributes to the limited ability of immigrant women to search, understand and use health-related information on the internet. Immigrants with older age and lower educational attainment have more problems accessing the internet and understanding and evaluating accurate and reliable health-related information on the internet. In a study conducted with Syrian immigrants in Sweden, it was shown that immigrants with low educational levels had limited health literacy [2]. Providing education to individuals with chronic diseases positively affected/increased rational drug use and health literacy [49]. Prolonged length of stay, positive perception of social status and educational level of immigrants in the country of migration affect the level of health literacy [20]. Immigrants are at high risk of having limited health literacy. This plays an important role in achieving better health for themselves and their families [50]. In another meta-analysis, it was shown that providing accessible and reliable health information on the internet or in the media in simple and understandable language would contribute to improving individuals’ health literacy levels [51].

### Strengths and Limitations of the Research

Given Turkey’s significant standing in global migration statistics, research conducted on migrants within the country undeniably offers critical contributions to the literature. The present study was meticulously executed in an area densely populated by migrants, employing Arabic-speaking interpreters. This approach ensured a direct engagement with the migrants, allowing for a more authentic representation of their voices and experiences. Specifically, by focusing on this distinct and often hard-to-reach migrant group, our research aims to fill a palpable gap in the literature by centering on their subjective evaluations.

However, it is imperative to underscore certain limitations of our study. Conducting the research in a singular region may impose constraints on the generalizability of the findings to the broader migrant population in Turkey. Additionally, the involvement of interpreters, while invaluable, could potentially raise concerns about the accuracy and impartiality of the translated responses from the migrants. Moreover, as the study predominantly focuses on Arabic-speaking migrants, it does not encompass insights from migrants of other linguistic backgrounds.

## 4. Materials and Methods

### 4.1. Research Type and Research Population

A cross-sectional study was conducted. The population of the study consisted of Syrian immigrants over the age of 18 who applied to Sultanbeyli Strengthened Migrant Health Center. Sultanbeyli is the district with a total population of 358,201 and has the lowest socioeconomic level in Istanbul. Around 22,000 Syrian immigrants live in the district.

Strengthened Migrant Health Centers are organizations that provide primary health care services to Syrian refugees who have settled in Turkey. These centers are staffed by specialist physicians, general practitioners, dentists, allied health personnel, psychologists and social workers. The centers are mostly staffed by Syrian healthcare professionals. Therefore, there is no language barrier/problem [52]. There are 8 of these centers in Istanbul, 1 of which is located in Sultanbeyli district. All immigrants over the age of 18 who volunteered to participate in the study were included in the study without sampling.

### 4.2. Measurement Tools

For the study, a questionnaire was prepared based on the literature and consisted of three sections. The first part of the questionnaire consisted of statements evaluating sociodemographic characteristics and health status. The second section includes statements on rational drug use prepared according to the guidelines and guidelines in the literature. The third section includes the E-Health Literacy Scale Arabic form. The survey was conducted face-to-face with immigrants through Arabic-speaking interpreters.

### 4.3. Rational Drug and Antibiotic Use Survey

The rational drug use questionnaire was prepared based on the World Health Organization’s (WHO) public awareness survey on antibiotic resistance conducted in 6 different WHO regions in 2015, the rational drug use scale whose validity and reliability studies have been conducted in Turkey, and other sources in the literature. This section consists of statements aiming to obtain information about the rational drug use status and attitudes of immigrants [7,30,53]. The statements in the section are in a 5-point Likert type and consist of a total of 13 items. Each item has a response scale ranging from “Strongly Disagree” to “Strongly Agree”. The section also includes negative statements. The relevant statements were compiled in order to learn the level of knowledge of the participants about the use of medicines and antibiotics and to evaluate their attitudes. The items in the section provide a subjective assessment of the rational use of medicines and antibiotics by immigrants [7,30].

### 4.4. E-Health Literacy Scale (eHEALS)

The eHEALS was developed by Norman and Skinner in 2006 and aims to measure literacy skills useful in assessing the effects of strategies for delivering online information and applications [1,18]. The eHEALS consists of 8 items, and participants are asked to rate each item on a 5-point Likert scale (strongly disagree, disagree, undecided, agree, or strongly agree). Total scores range from 8 to 40, with higher scores indicating higher self-perceived eHealth [1,54]. eHEALS scores are divided into thresholds of inadequate (8–20 points), problematic (21–26 points) and adequate (27–40 points). The eHEALS Arabic validity and reliability study was conducted by Wangdahl et al. [19]. However, since the use of 3 thresholds in the Arabic eHEALS threatens the validity and reliability of the scale, the scale was divided into two: limited (insufficient + problematic = 8–26 points) and sufficient (27–40 points). In our study, a 2-point version of the scale was used to identify those with eHealth literacy problems [1,19]. eHEALS psychometric tests show that it is a valid and reliable instrument and has also been translated, adapted and validated in Arabic [2,54].

### 4.5. Statistical Analysis

For statistical analysis, the eHEALS was accepted as the dependent variable. Statistical Package for the Social Sciences (SPSS) Program version 26.0 was used for statistical analysis. Continuous variables were expressed as mean ± standard deviation (SD) and median. Categorical variables were expressed as numbers and percentages (%). Kolmogorov–Smirnov and Shapiro–Wilk tests were performed for normality analysis of the data and Skewness and Kurtosis values of the scales with *p* < 0.05 were analyzed. It was accepted that the values with Skewness and Kurtosis values between ±1.5 were normally distributed, and the values not between ±1.5 were not normally distributed.

Since the data in the research group did not show normal distribution, the Mann–Whitney U test and Kruskal–Wallis test were used in data analysis. Chi-square and Fisher’s exact tests were used to compare categorical variables between groups. Correlation (Spearman) analysis was used for the relationship between continuous variables. Logistic regression analysis was performed to predict the level of eHealth literacy according to the independent variables, model fits were evaluated, and the variables that contributed significantly to the model were examined. In statistical analyses, *p* ˂ 0.05 was considered significant.

### 4.6. Ethics Committee Permission

Ethics committee permission was obtained from the Istanbul Medipol University Non-Interventional Clinical Research Ethics Committee on 24 November 2022 with decision number 991. The individuals included in the study were asked to participate in the study after being informed about the research and permissions. A questionnaire was administered to individuals who agreed to participate in the study.

## 5. Conclusions

It was observed that Syrian immigrants have very good knowledge and attitudes about antibiotic supply and use. However, it was observed that their knowledge and attitudes regarding drug use, treatment compliance and herbal medicines were not sufficient. The eHealth literacy level of 80.3% of the immigrants in the research group was found to be limited (insufficient + problematic) and 19.7% was found to be sufficient. The eHEALS level of Syrian immigrants was found to be associated with being married, having a low income level, living in Turkey for a longer period of time, chronic disease, regular medication use and monthly out-of-pocket health expenditure. In addition, advanced age, low education level, female gender and regular medication use affected the low level of eHealth literacy.

Interventions targeting disadvantaged groups such as immigrants are very important in preventing infectious diseases, reducing treatment costs and monitoring chronic diseases. At this stage, health literacy interventions play a critical role. In today’s digital environment. eHealth literacy interventions for immigrants will help them access reliable health information online and make the right decisions about their health. In addition, health promotion interventions such as eHealth literacy will enable immigrants to care about their health and improve their quality of life.

The success of health policies will be enhanced if countries with a high concentration of immigrants plan and implement health services by taking into account immigrants’ needs, learning competencies, language problems, living conditions and sociocultural characteristics. eHealth literacy interventions for immigrants will facilitate the provision of health services and contribute to the safe access of immigrants to health services.

## Figures and Tables

**Table 1 antibiotics-12-01531-t001:** Sociodemographic characteristics, health and disease data of immigrants by gender.

Age (Mean ± SD)	39.19 ± 13.10	
	*n*	%
Age Group		
18–29 Years	155	28.6
30–39 Years	135	24.9
40–49 Years	114	21.0
Age ≥ 50 years	138	25.5
Marital Status		
Married	417	76.9
Single	78	14.4
Divorced/Widowed	47	8.7
Education Level		
Primary school and below	188	34.7
Middle School	67	12.4
High School	261	48.2
University	26	4.8
Income Status		
Income less than Expenditure	436	80.4
Income matches expenditure	106	19.6
Employment Status		
Yes	219	40.4
No	323	59.6
Life expectancy in Turkey		
0–4 Years	36	6.7
5–6 Years	159	29.3
7 Years and above	347	64.0
Number of People in Household		
1–2 people	23	4.2
3–4 people	132	24.4
5–6 people	213	39.3
7 or more people	174	32.1
Presence of a disabled person at home		
Yes	49	9.0
No	493	91.0
Chronic disease status		
Yes	198	36.5
No	344	63.5
Regular medication use		
Yes	330	60.9
No	212	39.1
Approach when you are sick		
Consulting a physician	472	87.1
I did not do anything for a while	57	10.5
Buying medicine at the pharmacy	3	0.6
Do not use medicines according to your own experience	10	1.8
Receipt of prescribed medication		
Free at the pharmacy	294	54.2
With co-payment from the pharmacy	241	44.5
With mobile payment from the pharmacy	7	1.3
Availability of the most recently used medicine		
Pharmacy	537	99.1
Internet	3	0.4
Residual from previous use	2	0.6
Out-of-pocket spending for health		
Yes	366	67.5
No	176	32.5
Average monthly out-of-pocket expenditure		
˂500 TL	456	84.1
500–1000 TL	72	13.3
˃1000 TL	14	2.6

TL: Turkish Lira.

**Table 2 antibiotics-12-01531-t002:** Characteristics of Rational Drug Use among immigrants.

	Strongly Disagree/Disagree	Not Sure	Strongly Agree/Agree
	*n* (%)	*n* (%)	*n* (%)
People should only use medicines when prescribed by a doctor.	1 (0.2)	10 (1.8)	532 (97.0)
People should not keep antibiotics in their homes and then use them for other illnesses.	12 (2.2)	22 (4.1)	508 (93.7)
Doctors should only prescribe antibiotics when needed.	2 (0.4)	19 (3.5)	521 (96.1)
It is not using too many medicines. but using enough medicines that helps us heal.	19 (3.5)	112 (20.7)	411 (75.8)
When we feel better during treatment. we can stop taking the medication.	55 (10.1)	211 (38.9)	276 (51.0)
There is no harm in recommending medication to relatives with similar complaints.	206 (38.0)	203 (37.5)	133 (24.6)
Herbal products can be used instead of medicines.	210 (38.7)	124 (22.9)	208 (38.3)
Consuming herbal products as much as desired does not harm health.	47 (8.9)	155 (28.6)	339 (62.5)
There is nothing wrong with taking medicines that we have used before and that have helped us to get better. when we have the same symptoms. or asking the doctor to prescribe them.	153 (28.2)	232 (42.8)	157 (29.0)
We can decide when. how and for how long we take the medicine.	199 (36.7)	240 (44.3)	103 (19.0)
Medicines can be used in the same amount for all age groups.	331 (61.0)	163 (30.1)	48 (8.9)
The duration of treatment for each drug is equal to each other.	369 (68.1)	145 (26.8)	28 (5.2)
More expensive drugs are more effective.	365 (67.4)	147 (27.1)	30 (5.5)

**Table 3 antibiotics-12-01531-t003:** eHealth Literacy Scale (eHEALS) Scores of immigrants.

eHealth Literacy Scale—Median (Min–Max)	Median: 21 (8–34)	
	*n*	%
Limited (insufficient-problematic) (8–26)	435	80.3
Adequate (27–40)	107	19.7

**Table 4 antibiotics-12-01531-t004:** Comparison of sociodemographic characteristics and health-disease status with eHEALS.

Feature	eHEALS		Test Statistic	*p*
Gender	*n*	Rank Mean		
Woman	283	263.62	U = 38.878	0.219
Male	259	280.11		
Age Group				
18–29 years old ^1^	155	267.14	X^2^ = 9.169	0.027 ^2–4^ *
30–39 years ^2^	135	296.50		
40–49 years ^3^	114	280.49		
≥50 years ^4^	138	244.51		
Marital Status				
Married ^1^	417	287.41	X^2^ = 10.106	0.000 ^1−2^ * ^1−3^ *
Single ^2^	78	233.21		
Divorced/Widowed ^3^	47	193.87		
Education Level				
Primary school and below	188	290.85	X^2^ = 6.984	0.072
Middle School	67	261.84		
High School	261	256.45		
University	26	307.58		
Income Status				
Income less than Expenditure	436	289.42	U = 15.295.5	0.000
Income matches expenditure	106	197.80		
Employment Status				
Yes	219	267.59	U = 36.225.5	0.630
No	323	274.15		
Life expectancy in Turkey				
0–4 Years ^1^	36	124.44	X^2^ = 42.419	0.000 ^1−2^ * ^1−3^ * ^2−3^ *
5–6 Years ^2^	159	252.88		
7 Years and above ^3^	347	295.29		
Number of People in household				
1–2 people	23	285.39	X^2^ = 2.442	0.486
3–4 people	132	254.83		
5–6 people	213	271.99		
7 or more people	174	281.71		
Presence of a disabled person at home				
Yes	49	237.88	U = 13.726	0.113
No	493	274.84		
Chronic disease status				
Yes	198	249.91	U = 38.381	0.014
No	344	283.93		
Regular medication use				
Yes	330	254.57	U = 40.568.5	0.002
No	212	297.86		
Approach when you are sick				
Consulting a physician ^1^	472	252.74	X^2^ = 54.143	0.000 ^1−2^ * ^1−4^ *
I did not do anything for a while ^2^	57	398.59		
Buying medicine at the pharmacy ^3^	3	309.50		
Using medicines according to your experience ^4^	10	421.05		
Receipt of prescribed medication				
Free at the pharmacy	294	281.14	X^2^ = 3.044	0.218
With co-payment from the pharmacy	241	258.78		
With mobile payment from the pharmacy	7	304.50		
Out-of-pocket health expenditures				
Yes	366	270.08	U = 32.727.5	0.760
No	176	274.45		
Amount of out-of-pocket health expenditure (monthly average)				
˂500 TL	456	287.65	X^2^ = 31.012	0.000
500–1000 TL	72	183.11		
˃1000 TL	14	200		

U: Mann–Whitney U. X^2:^ Kruskal–Wallis * Refers to paired groups where there is a significant statistical difference. ^1–4^ It refers to group numbers

**Table 5 antibiotics-12-01531-t005:** Regression analysis of socio-demographic characteristics and eHEALS.

Factor	B	df	*p*	O.R	95% CI
18–29 Years (Ref.)					
30–39 Years	−0.042	1	0.887	0.959	0.541–1.700
40–49 Years	0.426	1	0.246	1.531	0.746–3.145
Age ≥ 50 years	1.042	1	0.019	2.836	1.190–6.760
Male (Ref.)					
Woman	0.474	1	0.048	1.606	1.005–2.567
University (Ref.)					
High School	0.429	1	0.369	1.535	0.603–3.911
Middle School	0.944	1	0.071	2.571	0.921–7.181
Primary school and below	1.378	1	0.003	3.966	1.583–9.936
Chronic Disease (No)					
Yes	0.623	1	0.127	1.865	0.838–4.147
Regular medication use (No)					
Yes	−1.707	1	0.000	0.181	0.092–0.359

Ref: Reference Category. B: Regression Coefficient. df: Degrees of Freedom. *p*: Significance Level. O.R: Odds Ratio. CI: Confidence Interval. Hosmer–Lemeshow test *p* value: 0.078. Nagelkerke R^2^: 0.233.

## Data Availability

All datasets and analyses used throughout the study are available from the corresponding author upon reasonable request.

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
