# Peer review of "Rational Drug and Antibiotic Use Status, E-Health Literacy in Syrian Immigrants and Related Factors: A Cross-Sectional Study"

_antibiotics, 2023, doi:10.3390/antibiotics12101531_

Round 1

Reviewer 1 Report

Thank you very much for allowing me to review this manuscript. It shows very interesting results on e-health literacy among the Syrian immigrant population and the use of antibiotics.
The manuscript and its subject matter falls within the editorial line of antibiotics journal, however, I would like to make certain contributions in order to contribute to improving its quality.
1.- Introduction:
1.a.- The first sentence, in which the term "health" is defined, should contemplate the most current definition and reference it.
1.b.- In general, the introduction is too long, which makes it difficult to read. It should be restructured into 3 paragraphs, in which the topic under study is well focused, clearly generating the conceptual framework and thus guiding the main objective.
2.- Methodology:
2.a.- Line 105, the initial phrase that refers to the design, should be modified. Starting with "The study was planned.." does not seem the most appropriate to me. The study is completed, so the planning phase is already known. The sentence should be "A cross-sectional study was conducted.
2.b.- The phrase in lines 115-118 is repeated. delete one of them.
2.c.- Measurement instruments: Have they been validated on the study population? This should be clear in the text, since it greatly compromises the reliability of the data and the validity of the results.
2.d.- On line 158, p<.05 appeared, it should be corrected by p<0.05.
3.- Results:
3.a.- Table 1: In the "Age" row, the mean and SD are reported. What appears in parentheses? Range? Interquartile range? If age follows a normal distribution, it is sufficient to report as mean and SD.
3.b.- Table 1: Why is the table presented broken down by sex? Why is sex not presented as another independent variable? as well as age, marital status... Are analyzes grouped by sex carried out afterwards? This type of analysis does not appear anywhere in the manuscript.
3.c.- Table 4: What are the subscript numbers that appear in certain cells of the last column of this table? It must be explained
3.d.- Lines216-217, first sentence. That phrase is an assessment of the research team, it should not appear in the results section. If the research team wishes, they can discuss the usefulness of logistic regression in the discussion section.
3.e.- Regarding the logistic regression analysis, certain independent variables are analyzed in Table 5. From my point of view, the regression model should incorporate all the independent variables that show statistical significance in the bivariate analyzes shown in Table 4. If the research team decides not to include them, they should discuss why.
4.- Discussion:
4.a.- In the first paragraph of the discussion, the most outstanding results of the research should be reported, based on the main objective, and the explanation given to it. In the manuscript, the first paragraph is a repetition of the results, so it should be modified.
4.b.- The section on strengths and limitations should be improved. Both the strengths and limitations of the design and the results must be reported.
Thank you so much.

Reviewer 2 Report

  • A brief summary

The purpose of this study was to assess the factors that influence health literacy and rational drug use among Syrian immigrants. For the study, a questionnaire was prepared based on the literature and consisted of three sections (sociodemographic characteristics and health status; rational drug use; E-Health Literacy Scale Arabic form).

The strength of the study is given in Table 2.

I recommend the manuscript for publication after a minor revision.

  • General concept comments

Article:

Title: I think the title does not convey that one part of the survey is focused on the appropriate use of antibiotics. I propose to supplement in this regard.

The manuscript deals with a current issue, health literacy, and rational use of drugs among Syrian immigrants. The number of survey participants is adequate.

The authors evaluated the results considering the limitations.

  • Specific comments.

Line 75: 'Internet' - Capital letter is not required.

Tables: Each table needs aesthetic editing. We usually write the % value in parentheses after the case number (e.g. 15 (23.5%)). In addition to the average, there is usually a number for the narrowness of the data, e.g. SD.

Table 4.

Why is the % in the 3rd column?

This value can be calculated from Table 1.

It would be better to see the number, distribution and percentage of the 2 groups according to eHealath (8-26 points and 27-40 points).

What does Row Mean mean? Why is it in the table? How should it be interpreted? I think it could be deleted, since the authors used groups (8-26 points and 27-40 points).

The authors mixed the decimal point and the comma. (e. g., Table 4; p value vs. other numbers)

General questions to help guide your review report for research articles

  • Is the manuscript clear, relevant for the field and presented in a well-structured manner?

Yes,

  • Are the cited references current (mostly within the last 5 years)? Does it include an abnormal number of self-citations?

Yes; 40 reference are current;  ~ 75 %

self-citations: about 2-3

  • Is the manuscript scientifically sound and is the experimental design appropriate to test the hypothesis?

Yes,

  • Are the manuscript’s results reproducible based on the details given in the methods section?

Yes,

  • Are the figures/tables/images/schemes appropriate? Do they properly show the data? Are they easy to interpret and understand? Are the data interpreted appropriately and consistently throughout the manuscript? Please include details regarding the statistical analysis or data acquired from specific databases.

no.

Tables: Each table needs aesthetic editing. We usually write the % value in parentheses after the case number (e.g. 15 (23.5%)). In addition to the average, there is usually a number for the narrowness of the data, e.g. SD.

  • Are the conclusions consistent with the evidence and arguments presented?

Yes,

  • Please evaluate the ethics statements and data availability statements to ensure they are adequate.

They are adequate.

Round 2

Reviewer 1 Report

Many thanks to the authors for their words. You have carried out a very quick review and correction according to my contributions. This makes the job of peer review much easier.
You have provided a response letter that makes it much easier to follow up on the responses to my comments.
In this current version of your manuscript, the introduction has been greatly improved, allowing for a better understanding of the research focus and its theoretical framework. I believe that it has also improved the discussion, since it allows the results to be better contextualized.
Regarding the explanation provided to my doubts with the logistic regression, it now allows me to better understand why some of the independent variables that had been previously analyzed had not been introduced.
In general, the article, from my point of view, has increased in quality, and at this moment, it perfectly meets the standards of this journal.
Thank you very much for your kind words, which positively reinforce the selfless work of the reviewers.